# Quality of Life in Cancer Patients: The Modern Psycho-Oncologic Approach for Romania—A Review

Monica Licu [1] , Claudiu Gabriel Ionescu [1],* and Sorin Paun [2]

[1] Department of Medical Ethics, Carol Davila University of Medicine and Pharmacy, 050474 Bucharest, Romania; monica.licu@umfcd.ro
[2] Department of General Surgery, Carol Davila University of Medicine and Pharmacy, 050474 Bucharest, Romania; sorin.paun@umfcd.ro
* Correspondence: claudiu.ionescu@drd.umfcd.ro

**Abstract:** Quality of life (QOL) is an important indicator of human satisfaction and well-being. QOL is significantly and persistently affected for patients after a cancer diagnosis. Despite some evidence suggesting that psycho-oncologic interventions can provide lasting benefits, the inclusion of such interventions in cancer therapy is not universal. This article provides an overview of the known approaches to the evaluation of QOL in cancer patients and various interventions for improving patients' outcomes, with a focus on the eastern European regional and specific Romanian context. With a mortality rate above and cancer care performance below the EU average and unequally distributed, Romania urgently needs a national coordination program, which is discussed in our review, highlighting the main psychological tools needed for the assessment and the challenges involved in implementing the program. In the end, we suggest some directions for the future development of the psycho-oncologic approach in the context of social considerations, policy, and the unexpected financial challenges the nation provides.

**Keywords:** quality of life; health-related quality of life; psychosocial factors; psycho-oncology

## 1. Introduction

Quality of life (QOL), as an essential dimension of human existence, has always been accepted as the ultimate goal of healthcare [1,2]. However only in modern times has the debate progressed beyond philosophy and onto the implementation of specific interventions and measures directed at the health-related quality of life. It is generally accepted that the modern era of the "quality-of-life" concept debuted through the efforts of the World Health Organization (WHO) [2], which expanded the definition of health to mean "not only the absence of infirmity and disease, but also a state of physical, mental, and social well-being" [3].

The WHO definition not only put health in a more holistic context, but also laid the foundations for operationalizing quality of life as a genuine measure of well-being. This meant that the measures of health went beyond the established clinical or biological outcomes (symptoms, signs, and clinical events) and approached more personal domains (such as perception of the treatment, self-image, and distress caused by cancer) [4].

QOL is a multidimensional construct that consists of several domains, among which are physical, mental, and social ones [4]. While there is agreement among the many definitions of QOL, there are also differences. The WHO definition of QOL is "individuals' perceptions of their position in life in the context of the culture and value systems in which they live and in relation to their goals, expectations, standards and concerns". QOL is referred to as the individual's satisfaction with life and perception of well-being [5]. QOL is defined as the subjective perception of satisfaction or happiness in the relevant domains of the individual [6]. Revicki [7] suggests that QOL represents "a broad range of human experiences related to one's overall well-being". Other authors have looked

at a narrow definition that is more directly related to the health of the individual. For instance, Csaba et al. [8] recognize two fundamental components of QOL. The first is the ability to execute daily activities, referring to physical, psychological, and social well-being (individual's perception of their own condition). The second component refers to the degree of satisfaction regarding one's level of functionality and control capacity in symptom evaluation and treatment [9]. The latter component, which health research usually refers to as "health-related quality of life", is the focus of this review. In the interest of convenience, we will refer to this concept as QOL. When considering QOL specifically in cancer patients, some authors suggest that it describes a "patients' appraisal of and satisfaction with their current level of functioning compared to what they perceive to be possible or ideal" [10].

## 2. Materials and Methods

The studies included in the review consist of published English language peer-reviewed articles focused on examining current psycho-oncological approach guidelines and their relationship with the Romanian policy. We utilized the following databases: Google Scholar, Elicit, PsycINFO, PubMed, and Science Direct, and conducted electronic searches from March 2023 to April 2023, examining articles published at any time prior to 15 March 2023. The searches were conducted with terms related to psycho-oncology (i.e., "psycho-oncology", "oncology", "psychosocial", "psycho-oncology guidelines", "cancer patients", and "Romania") and quality of life (i.e., "life quality", "health-related quality of life", and "quality-of-life indicators"). The concept of "quality of life" and "Romanian psycho-oncology" were very scarcely mentioned throughout the literature; thus, we included broader concepts such as "eastern Europe psycho-oncology" and "eastern Europe quality of life in cancer patients" to attain a broader perspective of our primary objectives. All the published articles found using the above search terms and deemed to be related to the topic of focus were included based on their relevance. The search was performed according to the PRISMA extension for scoping reviews and the PRISMA-ScR checklist [11]. The first search yielded 76 hits, which resulted in 20 articles after removing duplicates and articles not related to Romanian policy. Their titles and abstracts were screened for adequacy. Twenty relevant articles were finally included in the review (Figure 1) [12]. All authors read the full texts of all the selected articles and agreed upon their inclusion in the review. No additional articles were added after reviewing the references for the selected papers.

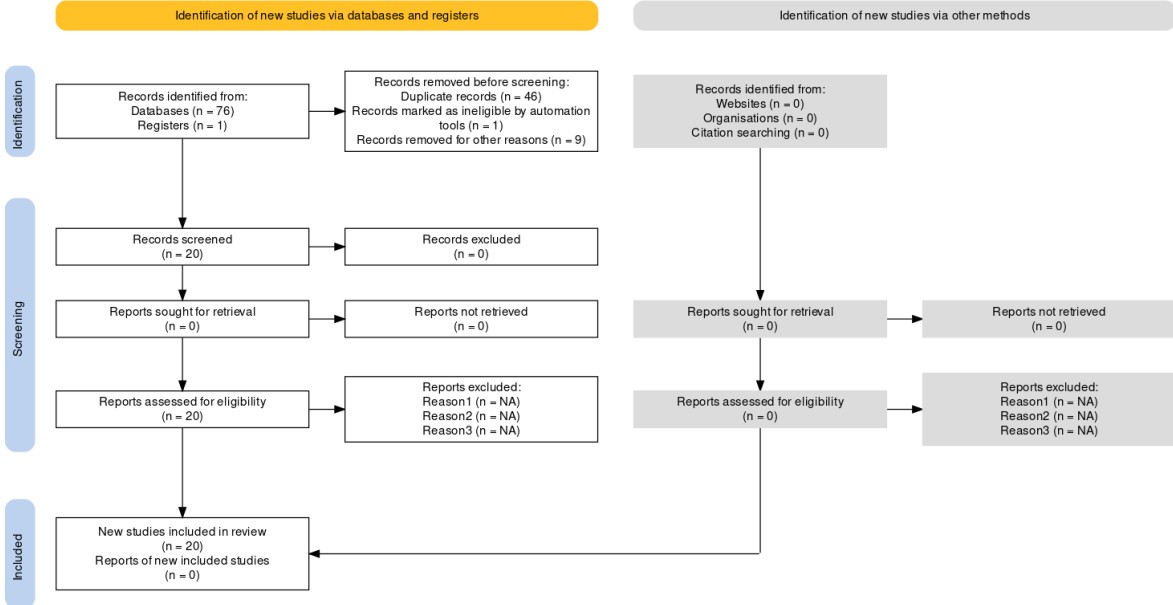

**Figure 1.** Flow diagram of included articles.

## 3. Considerations and Models of Health-Related Quality of Life

To map out existing conceptual relationships within the elements of QOL, a conceptual model was proposed, a version of which is presented in Figure 2 [13]. The novelty of this model is that it connects two aspects of research, clinical and social, and it provides a visualization of the biological, clinical, physical, psychological, and social factors influencing quality of life. This model has been adopted by many authors and adapted to better allow the targeting of specific QOL measurement tools [9].

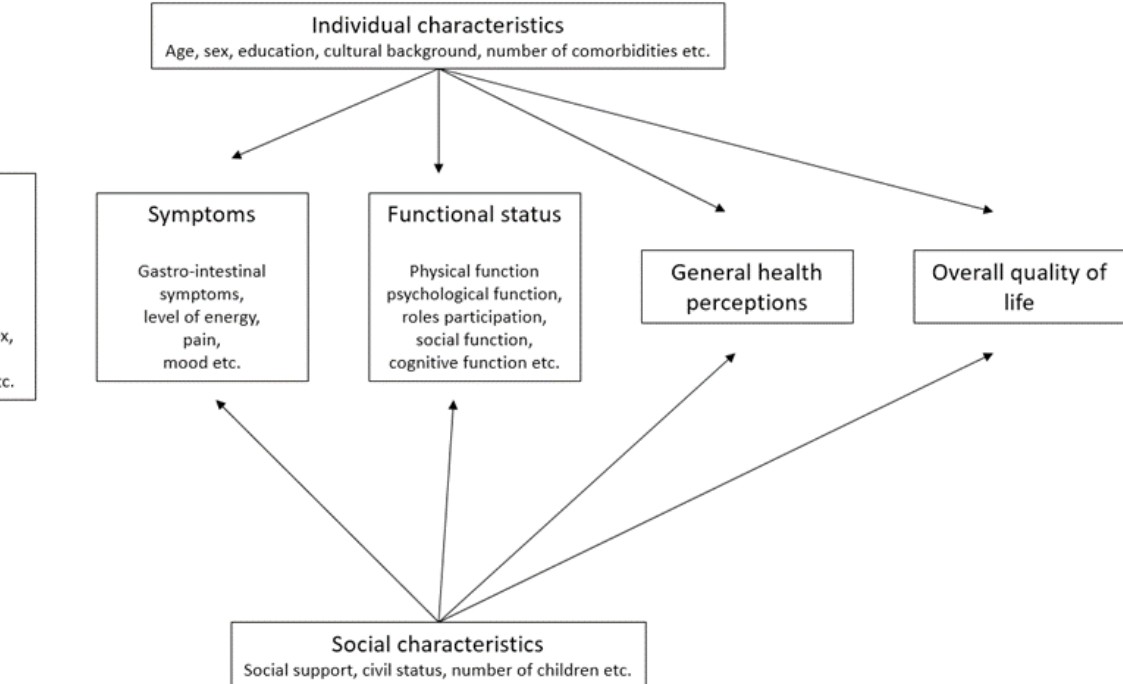

**Figure 2.** Conceptual model of factors involved in the health-related quality of life [13].

## 4. The Need to Assess QOL in Cancer Patients

In addition, QOL is a critical component of healthcare, as "the ultimate purpose of all health interventions is to enhance the quality of life" [14]. Consequently, a QOL assessment should be considered in all evaluations of health interventions.

The issue that Jenkins (and many other authors since) raises is that, although "people are the best experts on themselves" [15] and patients' perceptions are valid signs of QOL, when assessing QOL, it is necessary to rely as much as possible on observable phenomena and objective psychometric instruments, rather than subjective self-ratings. This is because much of the subjective perception of health is based on the patient's expectations (e.g., a patient not expecting a full recovery may exhibit a high satisfaction for only partial satisfaction) [16].

The English National Health Service (NHS) conducts an ongoing survey on the QOL of patients diagnosed with cancer [17]. Its aim is to explore the QOL changes for cancer patients in order to assess the service provision. The data collected as part of the survey help to better understand unmet needs. The survey results are reported in an interactive format and encompass several aspects of QOL, including health (such as mobility), functional domains (such as emotions) and symptoms (such as fatigue) [16]. In the September 2022 update, the survey reported that cancer patients reported a mean QOL level measured on the EQ-5D-5L index (EuroQol Research Foundation with five dimensions and three levels of severity) of 74.3, which was significantly lower ($p < 0.001$) than the level of 81.8 reported for the general population using the same

measurement instrument [17]. Cancer patients also reported significantly more problems than the general population for all aspects of health ($p < 0.001$ for each aspect). This difference was largest for usual activities. A total of 48.5% of cancer patients reported having mental health problems, compared with 33.1 in the general population. The cancer QOL survey [16] is currently one of the most important tools used to influence health policies and professional practice, and to enable patient empowerment in the United Kingdom.

QOL assessments are receiving more emphasis all around the globe. The International Psycho-Oncology Society (IPOS) functions as a multidisciplinary organization dedicated to promoting evidence-based approaches in psychosocial and behavioral oncology for the improvement of care in cancer patients [18]. The most recent IPOS standard of care, supported by many organizations worldwide, includes three core principles: (1) psychosocial cancer care should be recognized as a universal human right; (2) quality cancer care must integrate the psychosocial domain into routine care; and (3) distress should be measured as the sixth vital sign [18].

It has long been argued that the impact on QOL should be an important outcome to include in all clinical trials of health interventions, even as early as the regulatory step [19]. However, this has proven to be more difficult to implement in practice, due to many reasons presented, but mainly due to the lack of a universal QOL measure that is applicable in every clinical context.

Because of this methodological constraint, the United States Food and Drug Administration explicitly accepts surrogate outcomes instead of QOL outcomes when evaluating anti-cancer treatments [20], while the European Medicines Agency (EMA) advises care in using QOL outcomes and, more generally, patient-reported outcomes. In addition, even when the QOL measuring instruments are sufficiently validated, there is a concern that not all relevant domains are appropriately reported [21].

### 4.1. The Regional Context

In 2021, the European Union (EU) launched Europe's Beating Cancer Plan [22], a new EU-wide approach for the prevention, treatment, and care of cancer, focusing on areas where the European perspective would add the most value. This plan was structured around 10 key European initiatives to tackle all parts of the disease pathway. The European Commission allocated around EUR 4 billion for these actions, from sources that included the EU4Health program, Horizon Europe, and the Digital Europe program. This includes prevention, early detection, and improved diagnoses and treatments for cancer patients across Europe.

One of the actions specifically aims to improve the quality of life for cancer patients and survivors. Entitled the "better life for cancer patients initiative", this initiative will focus on follow-up care. The aim is to provide a "cancer survivor smart-card" that can summarize patients' clinical history and facilitate and monitor their follow-up care. A European Cancer Patient Digital Centre will also be launched to support the exchange of patients' data and the monitoring of survivors' health conditions [22].

Another key initiative of the cancer plan is the European Cancer Inequalities Registry. The aim of this registry is to provide reliable data on the disparities between EU member states and regions [22]. Part of this initiative was the drawing up of country cancer profiles in collaboration with the Organization for Economic Co-operation and Development (OECD) [23].

### 4.2. The Romanian Context

The Romanian cancer profile [23] reports a cancer mortality above the EU average, which has increased for six types of cancer since 2000. Authors attribute this to the possibly suboptimal performance in screening and early detection as well as "the scarcity of spe-

cialized professionals, uneven distribution of diagnostic and treatment facilities across the country, and lack of uniformity in following the protocols" [23].

At the end of 2022, the Cancer Prevention and Treatment Law [24] was adopted by the Romanian parliament. This law laid out the National Cancer Plan for 2023–2030, the current main policy document for all national actions related to cancer prevention and treatment. The plan specifically mentions the health services that Romanian citizens are entitled to. These include the prevention, diagnosis, and treatment of cancer; cancer care, including palliative care; psycho-oncology, cancer nutrition, and fertility services; and social assistance. More of the plan's details and actions will be set out in a regulation document, which is imminently being drafted by the Romanian government.

Within general aim 4 of the plan, cancer care, one of the current challenges is acknowledged: Romania is confronted with a critical deficit of psycho-oncology services, as well as professionals. There is currently no standard for the specific counselling of cancer patients regarding their feelings. As such, the plan recommends a standard for psycho-oncologic counselling of cancer patients and their families and monitoring of their quality of life. The country profile for Romania [23] also places the cancer care performance below the EU average and indicates that it is unequally distributed.

As an answer to these challenges, a National Recovery and Resilience Plan [25] was enacted to support policy and infrastructure reform. Disease registries were also planned, including for cancer, as a priority domain. This creates an opportunity to look at integrating psycho-oncologic services into the service package for cancer patients. Psychosocial factors play an important role in the evolution of cancer and can have a favorable or negative impact on disease outcomes [8]. According to Csaba [8], psycho-oncology, as a multidisciplinary research and practice field, is still in its infancy in Romania.

## 5. Measuring Quality of Life in Cancer Patients—Tools and Psychological Interventions

### 5.1. Psychological Tools

Several scales have been developed for the measurement of the impact of healthcare on QOL. Developing a scale that can accurately measure the QOL of cancer patients is a long and complicated process. Most scales are presented as questionnaires. Many QOL scales are specific, applying to certain conditions, populations, or functional issues. Others are generic. The generic tools are more useful when measuring QOL in people with comorbidities or when evaluating complex interventions [26]. Some generic scales include health profiles, which generate scores in several domains, and health utility measures, which generate a single score of QOL such as a quality-adjusted life-year [26].

One such generic questionnaire is the medical outcome study, a 36-item short-form health survey (SF-36) [27]. Composed of 36 items, it allows for an evaluation of the QOL of any population, and is related to eight domains (physical functioning, physical role, bodily pain, general health, vitality, social functioning, emotional role, and mental health).

Among the population-specific questionnaires, the quality-of-life questionnaire—core 30 (QLQ-C30) [28], developed by the European Organization for Research and Treatment of Cancer (EORTC), includes five function domains (physical, emotional, social, role, and cognitive), eight symptoms (fatigue, pain, nausea/vomiting, constipation, diarrhea, insomnia, dyspnea, and appetite loss), and global health/quality of life and financial impact [18]. Another widely used questionnaire is the functional assessment of cancer therapy—general (FACT-G) [29]; the FACT-G explores four domains of QOL in cancer patients: physical, social, emotional, and functional well-being.

A few site-specific cancer scales have been developed by the EORTC [30,31]. Among these are the QLQ-H&N35 [32] for head and neck cancers, the QLQ-BR23 [33] for breast cancer, and the QLQ-OV28 [34] for ovarian cancer. Some questionnaires evaluate a sin-

gle symptom; for instance, the functional assessment of cancer therapy—anemia (FACT-An) [29] was developed specifically for patients with anemia. The FACT-An is composed of the FACT-G, the fatigue subscale, and seven additional miscellaneous non-fatigue items relevant to anemia in cancer patients.

In 2014, the quality-of-life inventory (QOLI®) questionnaire [35] was validated for Romania. The QOLI was developed as a measure of well-being, positive health, and meaning. Its results are generally easy to understand and immediately suggest areas for intervention [35]. Other Romanian-validated instruments that can be further used for assessing quality of life are the SF-36 (short-form health survey) [36], the EQ-5D-3L (EuroQOL—five dimensions, three levels), the 5L, 5D instrument [37,38], and KIDSCREEN-27 [39], which have been applied in several studies. In our review, we found 20 studies in Romania assessing distress, mental well-being, fear of progression, coping, irrational beliefs, and emotion thermometers [40–43]. Regarding Romanian cancer patients, the QLQ-C30 versions of the EORTC, FACT-B, and SF-36 were used to assess quality of life in five cross-sectional studies with a focus on breast cancer, laryngeal cancer, and other psychological variables [40,44–47], as seen in Table 1.

**Table 1.** Studies assessing quality of life in cancer patients in Romania.

| Type of Study | No. of Participants | Quality of Life Used | Region | Year | Outcomes ($p < 0.05$) |
|---|---|---|---|---|---|
| Cross-sectional [44] | 420 | FACT-G (functional assessment of cancer therapy) | Transylvania | 2013 | Depression, anxiety, vital exhaustion, hopelessness, and illness intrusiveness were significantly and negatively correlated with quality of life, while sense of coherence was positively correlated with it. |
| Cross-sectional [45] | 23 | SF-36 (short-form 36) | Moldova | 2015 | The strongest correlation with QoL was found between the variable "cut down the amount of time spent on work and other activities" as a result of "physical health" and "limited in kind of work or other activities". |
| Cross-sectional [46] | 51 | FACT-B (functional assessment of cancer therapy) | Transylvania | 2015 | Emotional distress and a catastrophizing coping strategy had a negative effect on the QoL. |
| Cross-sectional [47] | 80 | QLQ-C30 | Transylvania | 2015 | There was a low score for the total laryngectomy group regarding functional scales: role, emotional, and social, and a high score for insomnia and financial difficulties. |
| Cross-sectional [40] | 330 | QLQ-C30 | Transylvania | 2021 | A loss of independence produced significant differences with large effect sizes in all the dimensions of QoL. |

Another argument for using the QOLI is the measure of life satisfaction across a number of areas said to comprise human happiness and contentment (after considering genetic contributions). These sixteen areas of life are: goals and values, self-esteem, health, relationships (in four areas: friends, love, children, and relatives), work and retirement,

play, helping or service, learning, creativity, money or standard of living, and surroundings (home, neighborhood, and community). Respondents rate how important each of the 16 domains is to their overall happiness and satisfaction.

The usefulness of the QOLI relies on the possibility of both planning and evaluating the benefit of both clinical and psychological interventions [48]. The QOLI is sensitive to change in studies of positive psychology coaching [49] and in studies of psychotherapy and medication effectiveness [35,48,49].

### 5.2. Psycho-Oncologic Interventions and Their Effect on the Quality of Life of Patients with Cancer

Many psychological interventions have been proposed for improving the QOL of patients with cancer. Such interventions range from short, nurse-led relaxation sessions to hour-long individual or group sessions led by a specially trained therapist [38]. In their 2002 systematic review, when looking at these interventions, Newell et al. [50] observed that those interventions involving structured or unstructured counselling as well as guided imagery seem to be effective for improving patients' general functional ability or quality of life. Follow-up evidence also suggested that the benefits would persist mid- to long-term [50].

The persistence of long-term effects from psycho-oncologic interventions was confirmed in a more recent meta-analysis by Faller et al. [51]. According to their meta-analyses, interventions such as individual psychotherapy, group psychotherapy, psychoeducation, and relaxation training produced small-to-medium effects on emotional distress, anxiety, depression, and QOL. The interventions seemed to be the most effective for patients with increased distress levels [51].

### 5.3. Structure of a Psycho-Oncologic Intervention

Psycho-oncologic interventions can be heterogeneous in their concept, structure, and duration. We will, however, use an existing intervention as an example of current developments in psycho-oncology practice.

Quality-of-life therapy (QOLT) was developed as a companion to the QOLI, building on the evidence gathered as part of the psychometric evaluation of the QOLI [15]. QOLT is a comprehensive, individually tailored package of positive psychology interventions suitable for both coaching and use in clinical contexts [13]. QOLT includes techniques for the control of negative feelings, making it a good match for the psycho-oncology context. QOLT patients are taught strategies and skills to help them identify, pursue, and fulfill their most cherished needs [52].

According to QOLT theory, a person's satisfaction with a particular area of life is made up of four parts: "(1) the objective characteristics or circumstances of an area, (2) how a person perceives and interprets an area's circumstances, (3) the person's evaluation of fulfilment in an area based on the application of standards of fulfilment or achievement, and (4) the value or importance a person places on an area regarding his or her overall happiness or well-being" [13]. This model is described by the acronym "CASIO" ("characteristics", "attitude about", "standards of fulfilment", and "importance"), where the last letter represents "overall satisfaction". QOLT always involves a two-track approach in which core techniques are combined with evidence-based elements of cognitive therapy.

## 6. Discussion

The evaluation of QOL in cancer patients is increasingly considered a mandatory step in this complex health intervention. Measuring the QOL not only reflects the impact of the medical and pharmacological treatment, but also the general well-being of the patient; this can suggest further areas of psycho-oncologic intervention. Unfortunately, this emerging practice is still in an early stage in Romania.

A cancer diagnosis has a long-term impact on the QOL of patients [53]. However, much of the interaction between patients and oncologists is focused on physical symptoms and clinical signs, while general well-being and QOL are perceived as lesser outcomes that would improve simply by following the prescribed therapy. However, the psychological factor is important for how patients cope with the disease and may represent an untapped resource for reducing patients' distress [54]. The inclusion of the psycho-oncologist as part of the mixed team can facilitate the treatment process by changing the focus to more patient-relevant outcomes. In addition, easy access to primary care facilities, which may include psycho-oncology experts, may improve the QOL of both patients and caregivers even more. Information centers, education facilities, and caregiver support groups may be points of interest located at the primary level of care such as GP offices or policlinics.

There are methodological and epistemic concerns surrounding the concept of QOL. In particular, the measurement of QOL is subject to considerable methodological uncertainty, due to the necessary extrapolation of subjective outcomes into objective measurements. Currently, despite significant practice and evidence building up over time, there are no universally applicable measurement tools for QOL, with the acceptable exception being Cantril's ladder scale, which has been used in general circulation and was developed with comprehensive attention to individual and diverse differences [55].

Another current challenge is the heterogeneity of QOL research in cancer. Some areas (e.g., breast cancer [54] and colorectal cancer [56,57]) are disproportionately more investigated than others (e.g., male patients with cancer [51]). More research on these aspects should provide a better understanding of the areas most sensitive to QOL interventions.

Finally, psycho-oncologic interventions take many forms, and information on their effectiveness is still limited [50]; the evidence does seem to suggest that patients with the worst QOL indicators seem to benefit the most, and the benefits seem to persist long-term [51]. These types of interventions also improve the quality of treatment and increase patients' quality of living and dying. In Romania, psycho-oncologists already work in some wards and clinics, especially in the major cities. They provide care for both the patient and their families. Associations, self-help groups, and therapeutic groups have also formed, but still represent isolated glimpses of hope across the country, which is experiencing growing unequal economic and social development between major cities and countryside areas. Unfortunately, psycho-oncologists are still a small group. The reimbursement of benefits under national health insurance is not enough for all those in need, and taking into account the epidemiological statistics previously presented, it should be assumed that the need for psycho-oncologists will significantly increase in the future in our country.

In the world, telemedicine, including telepsychiatry and tele-psychotherapy, is continually developing. In Romania, activities in these fields are still not represented enough. One such potential activity is the possibility of creating an online platform of psycho-oncological support. This project aims to promote psycho-oncology and create conditions for providing modern and effective psycho-oncological support to all patients, not depending on their location. It should be a useful tool for psychiatrists, psycho-oncologists, and other specialists. Another innovative solution may be the so-called online counseling of patients who are physically invalidated or those who wish to remain anonymous. Taking into account all the potential in terms of the development of Romanian psycho-oncology, the best has yet to come.

Psycho-oncological expertise can be applied worldwide and should be offered beyond race, culture, and socioeconomic status; however, it is necessary to consider the socio-cultural specificity of each country. Considering the major social and cultural variety of Romanian society, comprising many different cultural groups across the nation, a big challenge will be a so-called "Romanian model of distress" management plan, which may

be purposed by the authors. The Romanian Cancer Patients Associations, which are few, as the only professional health- and social-oriented groups of psycho-oncology in Romania, should strengthen their cooperation with the International Psycho-Oncology Society (IPOS) and other academic communities in Europe and the world. International cooperation with many neighboring European experts who share similar cultural and healthcare systems would help with the promotion of psycho-oncology in Romania.

In addition, further development cannot be made without training experts in the field and specialists that can provide care across the country. Conferences and training organized mostly by universities play an important role in the development of psycho-oncology in Romania. The leading medical universities, Bucharest and Cluj-Napoca, already hold post-university courses for physicians. The Romanian Psycho-Oncological Conference, dedicated to both physicians and psychologists, is held every year, with 2023 marking the third edition. Each edition has a main theme. In 2022, the conference was focused on the subject of the "multidisciplinary approach of pulmonary and breast cancer", with national and international guest speakers. As an addition to the above activities, essential for the development of psycho-oncology are the activities in the field of prevention and health promotion, undertaken mainly by NGOs, especially associations of oncological patients. Very prominent is the "restart to life" Cancer Patients Association, which is the voice of cancer patients in the entire regions of Muntenia and Dobrogea (psychological intervention groups held in 13 out of 41 counties of Romania). In addition, in 2020, a local project called CANPRIM was launched in Transylvania that focuses on the primary care approach to cancer distress [58]. The NGO works to improve the situation of all oncological patients and their caregivers. They also conduct nationwide preventive actions and offer training in psycho-oncology for psychologists. Future studies need to be held concerning how patients, caregivers, and healthcare professionals approach psychological well-being, cancer adaptation, coping strategies, and burnout. In addition, somatic and mental symptoms and their burden could represent an important topic for QOL and mental adaptations to cancer.

## 7. Conclusions

On a national level, the new health policy developments in Romania should open the way for better coordination between clinical and psychological evaluations and interventions in patients with cancer, with the potential to improve the QOL and well-being of such patients in order for the country to better adapt to the EU requirements for improving patients' and caregivers' cancer experience and overall quality of life. Romania will have access to the financial tools in order to attain this national objective, which is much-needed to be fulfilled. After our literature review and finding few studies concerning an assessment of the quality of life in cancer patients in Romania, we found that the QOLI is mandatory in the psycho-oncological approach, not only for assessments and data gathering, but also as an indicator of the disease prognosis and as mandatory help for health interventions.

**Author Contributions:** C.G.I. and M.L. contributed to the conception, structure of the paper, analysis, and interpretation of the available literature. C.G.I. and S.P. contributed to the development of the initial draft. M.L. reviewed and critiqued the output for important intellectual content. All the authors contributed to the article and approved the submitted version. All authors have read and agreed to the published version of the manuscript.

**Funding:** This research received no external funding.

**Informed Consent Statement:** Not applicable.

**Conflicts of Interest:** The authors declare no conflict of interest.

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
