# Peer review of "Quality of Life in Cancer Patients: The Modern Psycho-Oncologic Approach for Romania—A Review"

_curroncol, doi:10.3390/curroncol30070504_

Round 1

Reviewer 1 Report

I‘m thankful for the opportunity yo review your work.

The paper is interesting and even the topic.

Psycho-oncological approach could be reinforced not only by general approach, but dealing with the emerging primary setting fields.

The discussion paragraph is well written: I suggest to enhance the clinical perspective herewith suggested.

In conclusion, the paper is good opportunity for National growth in psychological interventions.

Author Response

Dear Reviewer,

Firstly, we thank you so much for taking your valuable time and helping us making our research more clearly defined and more meaning-oriented. We looked seriously and carefully at your comments and tried our best to improve the paper`s overall quality, as following:

Hint: We put the quotation from the manuscript on each point in order for you to easily identify our changes (turquoise color) and their specific lines in the manuscript.

Response to the main topics: „Psycho-oncological approach could be reinforced not only by general approach but dealing with the emerging primary setting fields.

The discussion paragraph is well written: I suggest to enhance the clinical perspective herewith suggested.

In conclusion, the paper is good opportunity for National growth in psychological interventions.”

  • Yes, the emerging primary fields should become a focus for our national program and we will try to lobby for that in order for the patients to have an easy access to their primary care provider while having access to information, education facilities and more also for the caregivers. Also, we added a brief discussion on the clinical perspective, current education and NGOs and current situation in Romania, in order to have a better current situation thus planning for a National growth (Lines 286-332)
  • We will discuss the challenge of clinical perspectives in the Discussions part as seen: „Also, the easy access to primary care facilities which may include psycho-oncology ex-perts may ease even more the QOL of both patients and caregivers. Information cen-ters, education facilities and caregivers support groups may be a point of interest locat-ed at the primary level of care like GP offices or policlinics. ”(Lines 267-270)

Thanks again for your valuable feedback that is helping us gain experience and perseverence towards better quality on our paper!

Best,

Authors

Reviewer 2 Report

Thank you for the chance to read your manuscript. This is a very interesting topic. The manuscript is well written for the most part. However, there are some significant changes that sound be made in order to improve its quality and scientific soundness. Firstly, the materials and methods section should be more concise. Some of the information should be relocated in a Table and in a Flow-Chart. I recommend explaining the total number of studies analyzed and the characteristics of the studies (e.g., observational study, randomized control study, cohort study). It would be useful to include another table to describe the overall psychological tools found in the scientific articles examined.

Finally, the discussion and conclusions section need to be implemented with further studies.

Author Response

Dear Reviewer,

Firstly, we thank you so much for taking your valuable time and helping us making our research more clearly defined and more meaning-oriented. We looked seriously and carefully at your comments and tried our best to improve the paper`s overall quality, as following:

Hint: We put the quotation from the manuscript on each point in order for you to easily identify our changes (turquoise color) and their specific lines in the manuscript.

Response to the main topics: „Firstly, the materials and methods section should be more concise. Some of the information should be relocated in a Table and in a Flow-Chart.

  • Yes, we certainly did not emphasize enough the materials and methods section and we apologize for that. After revisiting this part, we highlighted the guidelines used and created a flow diagram with the help of „Haddaway, N. R., Page, M. J., Pritchard, C. C., & McGuinness, L. A. (2022). PRISMA2020: An R package and Shiny app for producing PRISMA 2020-compliant flow diagrams, with interactivity for optimised digital transparency and Open Synthesis Campbell Systematic Reviews, 18, e1230. Here you have the quotation and the Figure 1 Flow chart of articles included: „ The search was performed according to the PRISMA extension for scoping reviews and the PRISMA-ScR checklist [11]. The first search yielded 74 hits, which resulted in 20 articles after removing duplicates and articles not related to Romanian policy. Their titles and abstracts were screened for adequacy. Twenty relevant articles were finally included in the review (Figure 1) [12]. All authors read the full-texts of all the selected articles and agreed upon their inclusion in the review. No additional articles were added after reviewing the references for the selected papers.” (Lines 72-83) ).

Response to „I recommend explaining the total number of studies analyzed and the characteristics of the studies (e.g., observational study, randomized control study, cohort study It would be useful to include another table to describe the overall psychological tools found in the scientific articles examined.”

  • Yes, we did not provide an overview of the tools dedicated to the assessment of quality of life in Romania. Our 20 articles found were mostly cross-sectional thus we didn`t emphasize that in a chart but, after this recommendation, we mentioned the studies concerning cancer patients as seen in the quotation and in Table 1: „ Other Romanian validated instruments which can be further used for assessing quality of life are SF-36 (Short-Form Health Survey) [36], EQ-5D-3L (EuroQoL-five-dimensions-3-level), 5D instrument [37] and 5L [38] and KIDSCREEN-27 [39] which were applied in several studies with only 5D being applied specifically in cancer patients. In our review we found studies in Romania assessing distress, mental wellbeing, fear of progression, coping, irrational beliefs, emotion thermometers[40-43] but no study focused on quality of life in cancer patients. Regarding Romanian cancer patients, the QLQ-C30 version of EORTC, FACT-B and SF-36 were used to assess quality of life in 5 cross-sectional studies with a focus on breast, laryngeal cancer and other psychological variables [44- 48] as seen in Table 1.” (Lines 221-250)

Type of study

No. of participants

Quality of life used

Region

Year

Outcomes

(P<.05)

Cross-sectional [44]

420

FACT-G (Functional Assessment of Cancer Therapy)

Transylvania

2013

depression, anxiety, vital exhaustion, hopelessness and illness intrusiveness were significantly and negatively correlated with quality of life, while sense of coherence was positively correlated with it

Cross-sectional [45]

23

SF-36

(Short Form-36)

Moldova

2015

The strongest correlation with QoL was found between the variable “cut down the amount of time spent on work and other activities” as a result of “physical health”  and “limited in kind of work or other activities”

Cross-sectional [46]

51

FACT-B (Functional Assessment of Cancer Therapy)

Transylvania

2015

Emotional distress and catastrophising coping strategy had a negative effect on the QoL

Cross-sectional [47]

80

QLQ-C30

Transylvania

2015

There was a low score in total laryngectomy group regarding functional scales: role, emotional and social and a high score on insomnia and financial difficulties

Cross-sectional [48]

330

QLQ -C30

Transylvania

2021

loss of independence produced significant differences with large effect sizes in all the dimensions of QoL.

Table 1. Studies assessing quality of life in cancer patients in Romania

Response to : „Finally, the discussion and conclusions section need to be implemented with further studies”

  • Yes, we added future points of interests in studying QoL and other psychological variables among cancer patients, caregivers and healthcare professionals as seen : „Future studies need to be held concerning how patients, caregivers and healthcare professionals present psychological wellbeing, cancer adaptation, coping strategies and burnout. Also, somatic and mental symptoms and their burden could represent an important topic on QoL and mental adaptation to cancer.” (Lines 403-406)

Thanks again for your valuable feedback that is helping us gain experience and perseverence towards better quality on our paper!

Best,

Authors

Reviewer 3 Report

I have two comments:

1. On line 100 it is stated: "mean QOL level 74.3", on line 101 it is stated: "than the 81.8 level". On what scale are those values?

2. I disagree with the statement on line 272: "there are no universally applicable measurement for QOL". The quality of life can be measured all over the globe on the Cantril scale of 0-10.

Author Response

Dear Reviewer,

Firstly, we thank you so much for taking your valuable time and helping us making our research more clearly defined and more meaning-oriented. We looked seriously and carefully at your comments and tried our best to improve the paper`s overall quality, as following:

Hint: We put the quotation from the manuscript on each point in order for you to easily identify our changes (turquoise color) and their specific lines in the uploaded manuscript.

Response to Point 1: On line 100 it is stated: "mean QOL level 74.3", on line 101 it is stated: "than the 81.8 level". On what scale are those values?

  • Yes, we did not mention the scale used in the British study. The scale used in the study was on EQ-5D-5L Index (EuroQol Research Foundation with 5 dimensions and 3 levels of severity). Here you have the quotation :„…mean QOL level measured on EQ-5D-5L Index (EuroQol Research Foundation with 5 dimensions and 3 levels of severity) of 74.3, significantly lower (p<0.001) than the 81.8 level reported in the general population using the same measurement instrument” (Lines 113-116)

Response to Point 2: I disagree with the statement on line 272: "there are no universally applicable measurements for QOL". The quality of life can be measured all over the globe on the Cantril scale of 0-10.

  • Yes, it is true and after revising the literature about the Cantril scale we found that it was an important point in the further development of other quality of life scales. We introduced the statement about the Cantril scale as seen in the quotation: „ measurement tools for QOL with the acceptable exception being Cantril's ladder scale which has been used in general circulation and has been developed with comprehen-sive attention to individual and diverse differences [41].” (Lines 345-347)

Thanks again for your valuable feedback that is helping us gain experience and perseverence towards better quality on our paper!

Best,

Authors

Reviewer 4 Report

Dear All,

Having analysed the paper entitled “Quality of life in cancer patients: the modern Psycho-Oncologic approach for Romania - a review”, I reached the following conclusions:

1.      The content of the work is consistent with the aim of the work.

2.      The work addresses an important issue related to oncological care.

3.      Although there are numerous publications devoted to the subject of quality of life and health-related quality of life, the paper manages to fill in the gaps in the already published works.

4.       The choice of methodology is not clear. The authors provide information on which databases they searched through and what criteria were used for searching but there is no detailed information on the results of the conducted analysis. The authors also fail to explain what approach was used for the analysis (e.g. PRISMA method).

5.      The Conclusion section does not discuss the results of databases’ search. I also believe that the authors did not provide information on which tools for assessing the quality of life from the group of general and dedicated tools for oncological patients have been validated for Romania (apart from the Quality of Life Inventory questionnaire).

6.      I suggest that the Discussion and conclusions section should be divided into two separate sections. The Discussion section needs to be further developed. It should focus on discussing the reasons behind the current state of oncological care and assessing the quality of life in Romania.

7.      The current structure of the paper does not include the Conclusions section.

8.      The research literature is out of date. The authors might want to consider refraining from referencing works published in 1958 and in the 90s.

Author Response

Dear Reviewer,

Firstly, we thank you so much for taking your valuable time and helping us making our research more clearly defined and more meaning-oriented. We looked seriously and carefully at your comments and tried our best to improve the paper`s overall quality, as following:

Hint: We put the quotation from the manuscript on each point in order for you to easily identify our changes (turquoise color) and their specific lines in the manuscript.

Response to Point 1, 2 and 3: The content of the work is consistent with the aim of the work. The work addresses an important issue related to oncological care. Although there are numerous publications devoted to the subject of quality of life and health-related quality of life, the paper manages to fill in the gaps in the already published works.

  • Thank you for highlighting our content is consistent with our aim which is a call for a national coordination and policy changes towards better access for cancer patients in high quality Psycho-Oncological interventions for our nation.

Response to Point 4:  The choice of methodology is not clear. The authors provide information on which databases they searched through and what criteria were used for searching but there is no detailed information on the results of the conducted analysis. The authors also fail to explain what approach was used for the analysis (e.g. PRISMA method).

  • Yes, we certainly did not emphasize the methodology choice and procedure and we apologize for that. After revisiting this part, we highlighted the guidelines used and created a flow diagram with the help of „ Haddaway, N. R., Page, M. J., Pritchard, C. C., & McGuinness, L. A. (2022). PRISMA2020: An R package and Shiny app for producing PRISMA 2020-compliant flow diagrams, with interactivity for optimised digital transparency and Open Synthesis Campbell Systematic Reviews, 18, e1230”. we found it was in line with our aims and recent literature. Here you have the quotation and the Figure 1 Flow chart of articles included: „ The search was performed according to the PRISMA extension for scoping reviews and the PRISMA-ScR checklist [11]. The first search yielded 74 hits, which resulted in 20 articles after removing duplicates and articles not related to Romanian policy. Their titles and abstracts were screened for adequacy. Twenty relevant articles were finally included in the review (Figure 1) [12]. All authors read the full-texts of all the selected articles and agreed upon their inclusion in the review. No additional articles were added after reviewing the references for the selected papers.” (Lines 72-83) and Figure 1 as seen below:

Response to Point 5: The Conclusion section does not discuss the results of databases’ search. I also believe that the authors did not provide information on which tools for assessing the quality of life from the group of general and dedicated tools for oncological patients have been validated for Romania (apart from the Quality of Life Inventory questionnaire).

  • Yes, we did not add a different Conclusions section and we proceeded to this. We added the section and we briefly highlighted the findings in our study and why there is need for an extensive review of current standards of practice in Romanian Psycho-Oncology as seen in quotation: „ After our literature review and finding few studies concerning assessment of quality of life in cancer patients in Romania, we found that QOLI is mandatory in the Psycho-Oncological approach not only for assessment and data gathering but also as an indicator of disease prognosis and help for health intervention.” (Lines 337-340)
  • Yes, we did not provide an overview of the tools dedicated to the assessment of quality of life in Romania. After a review of the studies and literature which we found very scarce, we found several scales applied and validated for multiple psychological variables, but no results on quality of life in cancer studies, and we mentioned them as seen: „ Other Romanian validated instruments which can be further used for assessing quality of life are SF-36 (Short-Form Health Survey) [36], EQ-5D-3L (EuroQoL-five-dimensions-3-level), 5D instrument [37] and 5L [38] and KIDSCREEN-27 [39] which were applied in several studies with only 5D being applied specifically in cancer patients. In our review we found studies in Romania assessing distress, mental wellbeing, fear of progression, coping, irrational beliefs, emotion thermometers[40-43] but no study focused on quality of life in cancer patients.” (Lines 221-228)

Response to Point 6: I suggest that the Discussion and conclusions section should be divided into two separate sections. The Discussion section needs to be further developed. It should focus on discussing the reasons behind the current state of oncological care and assessing the quality of life in Romania.

  • Yes, and we thank you for underlying this clarifying point. We proceeded to splitting the Discussions and Conclusions into two different sections.
  • Furthermore, we added extensive information about the current state of Psycho-Oncological care in Romania by highlighting the NGOs, education facilities and universities programs as seen: „ These types of interventions improve also the quality of treatment and also increases patients` quality of living and dying. In Romania, psycho-oncologists work already in some wards and clinics, especially in the major cities. They provide care for both the pa-tient and their families. Associations, self-help groups and therapeutic groups are also formed but still represent isolated glimpses of hope across the country which has a growing unequal economic and social development between major cities and country-sides areas. Unfortunately, psycho-oncologists are still a small group. Reimbursement of benefits under national health insurance is not enough for all in need and taking into account the epidemiological statistics previously presented, it should be assumed that the need for psycho-oncologists will significantly increase in the future in our country. In the world, telemedicine including telepsychiatry and tele-psychotherapy are further developing. In Romania activities in these fields are still not represented enough. One of such potential activity is the possibility of creating an Online Platform of Psycho-oncological support. This project aims at promoting psycho-oncology and creating conditions for providing modern and effective psycho-oncological support to all patients, not depending on their location. It should be a useful tool for psychiatrists, psycho-oncologists and other specialists. Another innovative solution may be the so-called online counseling for patients who are physically invalidated or those who wish to remain anonymous. Taking into account all the potentialities, in terms of development of Romanian Psycho-Oncology, the best has yet to come. Psycho-oncological expertise can be applied worldwide and should be offered beyond race, culture and socioeconomic status; however, it is necessary to consider the so-cio-cultural specificity of each country. Considering the major social and cultural variety of Romanian society, comprising many different cultural groups across the nation, a big challenge will be a so-called „Romanian Model of Distress” management plan which may be purposed by the authors. The Romanian Cancer Patients Associations, few, as the only professional health and social oriented groups of Psycho-Oncology in Romania, should strengthen the cooperation with the International Psycho-Oncology Society (IPOS) and other academic communities in Europe and the world. International cooperation with many neighboring European experts who share similar cultural and healthcare systems would help with the promotion of psycho-oncology in Romania.

Also, further development cannot be made without training experts in the field and specialists providing care across the country. Conferences and trainings organized mostly by universities play an important role in the development of Psycho-Oncology in Romania. The leading medical universities, Bucharest and Cluj-Napoca already held Post-University courses for physicians. The Romanian Psycho-Oncological Conference, dedicated to both physicians and psychologists, is held every year with 2023 marking the 3rd edition. Each edition had its main theme. In 2022 the conference was focused on the subject “Multidisciplinary approach of pulmonary and breast cancer” with national and international guest speakers. Addition to the above activities, essential for the development of Psycho-Oncology are the activities in the field of prevention and health promotion, undertaken mainly by NGOs, especially associations of oncological patients. Very prominent is “Restart to Life” Cancer Patients Association, who is the voice of cancer patients in the whole regions of Muntenia and Dobrogea (psychological intervention groups held in 13 out of 41 counties of Romania). Also, in 2020, a local project called CANPRIM was launched in Transylvania focused on primary care approach of cancer distress [52]. The NGO works to improve the situation of all oncological patients and their caregivers. They are also conducted nationwide preventive actions and ofer training in Psycho-Oncology for psychologists.” (Lines 355-402)

Response to Point 7: The current structure of the paper does not include the Conclusions section.

  • As mentioned above, we proceeded to including Conclusions section as a separate one which comprise the main findings of our review highlighting the need for urgent change in Romanian Psycho-Oncological program. (Lines 413-416)

Response to Point 8: The research literature is out of date. The authors might want to consider refraining from referencing works published in 1958 and in the 90s.

  • Yes, we understand this point and we thoroughly changed the outdated references (number 2,3,5,6,10,13,14,19,27-31 and 35) and searched for up-to-date studies pointing out our main ideas as seen in the new references highlighted in turquoise color in the manuscript.

Thanks again for your valuable feedback that is helping us gain experience and perseverence towards better quality of our paper!

Best,

Authors

Round 2

Reviewer 1 Report

The paper was worked efficiently and following all my suggestions

I think the paper is suitable for the Journal

Reviewer 2 Report

I suggest expanding and updating your references regarding the modern psycho-oncological approach. See for example these works:

DOI: 10.3389/fpsyg.2018.02487;

DOI: 10.1136/bmjopen-2022-063435;

DOI:  https://doi.org/10.3390/curroncol28040224

Reviewer 3 Report

I agree to publish the paper.

Reviewer 4 Report

I accept the revised version of the article.